# Methicillin-Resistant *Staphylococcus aureus* (MRSA) in a Tertiary Care Hospital in Kuwait: A Molecular and Genetic Analysis

**DOI:** 10.3390/microorganisms12010017

**Published:** 2023-12-21

**Authors:** Wadha A. Alfouzan, Samar S. Boswihi, Edet E. Udo

**Affiliations:** 1Microbiology Unit, Department of Laboratories, Farwaniya Hospital, P.O. Box 13373, Farwaniya 81004, Kuwait; 2Department of Microbiology, Faculty of Medicine, Kuwait University, P.O. Box 24923, Safat 13110, Kuwait; samar.boswihi@ku.edu.kw (S.S.B.); udo.ekpenyong@ku.edu.kw (E.E.U.)

**Keywords:** MRSA, *spa* typing, DNA microarray, molecular typing, antibiotic resistance

## Abstract

Methicillin-resistant *Staphylococcus aureus* (MRSA) is a major pathogen that causes serious infections in healthcare facilities and in communities. The purpose of this study was to investigate MRSA isolates obtained in a tertiary hospital in Kuwait to assess their antibiotic susceptibility profile and clonal composition. Sixty MRSA isolates collected in 2020 were tested through antibiotic susceptibility testing, *spa* typing, and DNA microarray analysis. All isolates were found to be susceptible to vancomycin (MIC: ≤3 µg/mL), teicoplanin (MIC: ≤3 µg/mL), rifampicin, and mupirocin, but were resistant to fusidic acid (n = 43, 72%), trimethoprim (n = 27, 45%), ciprofloxacin (n = 31, 51.7%), gentamicin (n = 14; 23.3%), kanamycin (n = 20; 33.3%), chloramphenicol (n = 7; 11.7%), tetracycline (n = 17; 28.3%), erythromycin (n = 19; 31.6%), inducible clindamycin (n = 13; 21.7%), and constitutive clindamycin (n = 2; 3.3%). The isolates belonged to 30 *spa* types and 13 clonal complexes (CCs). The dominant *spa* types were t304, t442, t311, t688, and t1234, collectively constituting 28.3% of the isolates. The dominant CCs were CC5 and CC6, which together constituted 46.7% of the isolates. This study provides updated research on antibiotic resistance and changes in the clonal composition of MRSA in a Kuwait hospital, including the disappearance of the ST239-MRSA-III clone that was previously the dominant clone in this hospital.

## 1. Introduction

Methicillin-resistant *Staphylococcus aureus* (MRSA) is a highly transmissible pathogen that spreads rapidly, causing outbreaks in healthcare systems. MRSA causes various types of infections that range from skin and soft tissue infections to invasive infections such as osteomyelitis, meningitis, pneumonia, and endocarditis [1,2]. Treating MRSA infections can be challenging because of the resistance of their bacteria to multiple antimicrobial agents [1]. This antibiotic resistance, particularly to the entire class of beta-lactam antibiotics, first emerged when MRSA strains acquired the *mecA* gene. The chromosomally located gene *mecA* encodes for penicillin-binding protein-2a (PB2a), which makes the pathogen avoid the inhibitory effects of beta-lactams [2].

MRSA isolates can be classified as hospital-acquired (HA-MRSA), community-acquired (CA-MRSA), or livestock-acquired MRSA (LA-MRSA) [2] on the basis of their antibiotic resistance, clinical presentations, and genomic characteristics [3]. HA-MRSA strains were first reported in 1961 in the UK among elderly patients who were admitted to hospitals and had received antibiotic treatments [3,4]. Most of these HA-MRSA isolates were resistant to multiple antibiotics and carried SCC*mec* types I, II, and III [3]. About 30 years later, another type of MRSA was reported among apparently healthy people in the community who had not been hospitalized and were not on any antibiotic treatments, and this was designated community-acquired MRSA (CA-MRSA) [5]. The CA-MRSA strains were first reported in Western Australia in 1993 [5], and later in other parts of the world [3]. CA-MRSA isolates were usually found to be susceptible to non-beta-lactam antibiotics, and carried SCC*mec* types IV, V, and VI. Then, in the early 2000s, new strains of MRSA associated with animal infections, designated livestock-acquired MRSA (LA-MRSA), started to appear in humans [6,7,8]. These LA-MRSA strains are also usually susceptible to non-beta-lactam antibiotics and carry SCC*mec* types IV and V, like the CA-MRSA strains.

Several MRSA lineages have been defined by using various molecular typing techniques, including pulsed-field gel electrophoresis (PFGE), multilocus sequence typing (MLST), staphylococcal protein A (*spa*) typing, DNA microarray, and whole genome sequencing [3,9], in order to monitor the spread of this pathogen in healthcare systems. In the last decade, changes in the distribution of MRSA lineages or clones have been reported in many countries [10,11,12,13]. These studies have demonstrated a decline in the prevalence of the pandemic HA-MRSA clone CC8/ST239, while the prevalence of CA-MRSA lineages have increased [3]. Some studies have also highlighted an increase in LA-MRSA among human patients in hospitals [3,14]. It is important to monitor these changes in order to control the spread of MRSA, which, in turn, will reduce the burden of MRSA infections in hospitals [15].

The constant changes in the MRSA population and lineages justify the continuous monitoring of MRSA to detect the introduction of novel lineages, provide appropriate treatment, and control their transmission [15]. Accordingly, changes were observed in an MRSA population that was isolated from patients in Kuwait hospitals between 1992 and 2010 [16]. During this period, CC8/ST239-MRSA-III, which was the dominant clone in the 1990s, declined gradually and was overtaken by different clones of CA-MRSA, including novel CA-MRSA clones [16]. MRSA isolates have been isolated from the Farwaniya hospital in Kuwait since the early 1990s. Molecular typing of MRSA isolates from the 1990s revealed the persistence and dominance of the HA-MRSA clone CC8/ST239-MRSA-III [16]. Another study, conducted in 2016 in a tertiary hospital, also revealed the dominance of this CC8/ST239 clone among MRSA isolates [17]. During the COVID-19 pandemic, the amount of MRSA isolated in this hospital declined significantly and results of sensitivity tests showed patterns that were different from those observed before COVID-19. This study examined MRSA isolates obtained during the COVID-19 pandemic in 2020 for their susceptibility to antibiotics and clonal composition.

## 2. Materials and Methods

### 2.1. Ethical Approvement

Ethical approval was not required for this study since no human subjects were involved.

### 2.2. MRSA Isolates

A total of 60 MRSA isolates were cultured from different clinical samples in 2020 at the Farwaniya hospital. The isolates were collected from various clinical specimens as part of routine diagnostic care and submitted for molecular typing at the MRSA Reference Laboratory located in the Department of Microbiology, College of Medicine, Kuwait University. The isolates were initially identified in the diagnostics lab of Farwaniya hospital using routine standard techniques including Gram strain, tube coagulase, and Vitek-2. COMPACT (BioMerieux, St. Louis, MO, USA). Once received in the MRSA Reference laboratory, the isolates were sub-cultured on brain–heart infusion agar (BHIA) and incubated at 35 °C for 18 h. Pure cultures were preserved in beads and stored at −20 and −80 °C. The isolates were recovered by sub-culturing twice on BHIA and incubated at 35 °C prior to testing.

### 2.3. Antimicrobial Susceptibility Profile of the MRSA Isolates

Clinical Laboratory Standard Institute (CLSI) guidelines were followed to perform antibiotic susceptibility testing using the Kirby–Bauer disc diffusion method [18] with various antibiotic discs (Oxoid). The following antibiotics were tested: penicillin G (2U), cefoxitin (30 µg), kanamycin (30 µg), mupirocin (200 µg), gentamicin (10 µg), erythromycin (15 µg), clindamycin (2 µg), chloramphenicol (30 µg), tetracycline (10 µg), trimethoprim (2.5 µg), fusidic acid (5 µg), rifampicin (5 µg), ciprofloxacin (5 µg), and linezolid (30 µg). The minimum inhibitory concentrations (MICs) for cefoxitin, mupirocin, vancomycin, and teicoplanin were determined with Etest strips (BioMerieux, Marcy-l′Étoile, France) according to the manufacturer’s instructions, and interpreted according the CLSI guidelines [18]. The breakpoints of the European Committee on Antimicrobial Susceptibility Testing were used to determine the susceptibility of the isolates to fusidic acid [19]. *S. aureus* strains ATCC25923 and ATCC29213 were used as a quality control strains for the disk diffusion and MIC determination, respectively.

### 2.4. DNA Extraction for PCR Testing

Prior to DNA extraction, the bacterial cells were grown on BHIA overnight at 35 °C. Three to five identical colonies of the overnight culture were picked using a sterile loop and suspended in a microfuge tube containing 50 μL of lysostaphin and 10 μL of RNase solution. The tube was incubated at 37 °C in a heating block (Thermomixer, Eppendorf, Hamburg, Germany) for 20 min. To each sample, 50 μL of proteinase K (20 mg/mL) and 150 μL of Tris buffer (0.1 M) was added and mixed by pipetting. The tube was then incubated at 60 °C in a water bath (VWR Scientific Co., Shellware lab, Radnor, PA, USA) for 10 min. The tube was then transferred to the heating block at 95 °C for 10 min in order to inactivate proteinase K activity. Finally, the tube was pulse centrifuged to remove any condensation on the lid. The extracted DNA was stored at 4 °C until it was used for PCR analysis.

### 2.5. Genotyping of the MRSA Isolates

All MRSA isolates were typed using staphylococcal protein A (*spa*) typing and DNA microarray methods.

### 2.6. Spa Typing

*Spa* gene amplification was performed using the synthetic primers published by Harmsen et al., [20]. The PCR protocol consisted of an initial denaturation (for 4 min at 94 °C), followed by 25 cycles of denaturation (1 min at 94 °C), annealing (1 min at 56 °C), and extension (3 min at 72 °C), and a final cycle with a single extension (5 min at 72 °C). Five µL of the PCR product was analyzed via agarose gel electrophoresis. Then, 20 μL of the PCR product was purified using an ExoSAP-IT kit according to the manufacturer’s protocol. In a sterile microfuge tube, 5 µL of the PCR product was mixed with 2 µL ExoSAP-IT™ reagent and placed in thermocycler machine (Bio-Rad, Hercules, CA, USA) at 37 °C for 15 min to degrade the primers and nucleotides, followed by 80 °C for 15 min to inactivate the ExoSAP-IT™ reagent. The purified DNA was then used for PCR sequencing. The PCR sequencing protocol consisted of an initial denaturation (for 1 min at 94 °C), followed by 25 cycles of denaturation (10 s at 96 °C), annealing (5 s at 55 °C), and extension (4 min at 66 °C). To remove the unused fluorescent dye deoxyterminator, a DyeEx (Qiagen, Germantown, MD, USA) kit was used according to the manufacturer’s protocol. The purified DNA was analyzed in an automated 3500 genetic analyzer (Applied Biosystems, Waltham, MA, USA) in accordance with the manufacturer’s protocol. The *spa* sequence was analyzed using the Ridom Staph Type software Version 2. 2. 1 (Ridom GmbH, Wurzburg, Germany). This software detected the *spa* repeats and assigned each isolate a *spa* type.

### 2.7. DNA Microarray Analysis

The DNA microarray was performed using INTER-ARRAY Genotyping Kit *S. aureus* (Inter-Array GmbH, Bad Langensalza, Germany) according to the manufacturer’s protocol and the method described previously in [21,22]. The microarray analysis was used to assign clonal complexes (CCs) and determine the carriage of genes responsible for antibiotic resistance and virulence factors.

## 3. Results

The MRSA isolates in this study were obtained from 60 patients, comprising 35 males (58%) and 25 females (42%), located at different wards in the Farwaniya hospital. Sixty-three percent of the isolates were obtained from blood (n = 16) and skin and soft tissue (n = 22) samples. The remaining isolates were obtained from urine (n = 7), endotracheal tracts (n = 4), ears (n = 2), high vaginal swabs (n = 2), eyes (n = 2), fluid (n = 1), fine-needle aspirate (n = 1), the groin (n = 1), nasal swab (n = 1), or sputum (n = 1).

### 3.1. Distribution of SCCmec Types, Spa Types, and Clonal Complexes among the MRSA Isolates

Thirty *spa* types were identified among the studied MRSA isolates. The distribution of these *spa* types is presented in Figure 1. The dominant *spa* types were t304 (n = 4), t442 (n = 4), t311 (n = 3), t688 (n = 3), and t1234 (n = 3). *Spa* types t701 and t5923 were each identified in two isolates. Twenty-three *spa* types occurred in single isolates. Sixteen isolates could not be assigned to a *spa* type.

Other *spa* types (detected in single isolate) included: t008, t019, t088, t104, t1090, t1120, t1192, t145, t1991, t232, t2336, t267, t3057, t315, t3594, t391, t450, t502, t5168, t639, t650, t682, and t757.

The SCC*mec* types inferred from the DNA microarray analysis were type V (n = 26) and type IV (n = 24). Three MRSA isolates carried both SCC*mec* types IV and V (SCC*mec* IV + V), and six isolates carried type VI.

Our DNA microarray analysis identified 13 clonal complexes (CCs) among the 60 MRSA isolates. The distribution of the CCs is summarized in Figure 2 and Table 1. The CC5 clone identified in 19 isolates (31.67%) was the dominant CC. This was followed by CC6 (n = 9), CC15 (n = 4), CC22 (n = 4), and CC97 (n = 4). The other CCs, CC30, CC80, CC88, CC121, and CC361, occurred in three isolates; CC8 and CC152 were each identified in two isolates; and CC1 was identified in one isolate.

The nineteen CC5 isolates were classified into five genotypes: CC5-MRSA-V + SCCfus, WAMRSA-14/109 (n = 10); CC5-MRSA-VI + SCCfus (n = 6); CC5-MRSA-IV [PVL+] (n = 1); CC5-MRSA-IV, pediatric clone [edinA+],WAMRSA-65 (n = 1); and CC5-MRSA-V [sed/j/r+],WAMRSA-11/34/35/90/108 (n = 1). The CC6 isolates belonged to two genotypes, consisting of CC6-MRSA-IV,WAMRSA-51 (n = 8) and CC6-MRSA-IV + V,WAMRSA-66 (n = 1).

All four CC15 isolates belonged to a single genotype: CC15-MRSA-V + SCCfus. The four CC22 isolates were identified as CC22-MRSA-IV [tst1+],UK-EMRSA-15/Middle Eastern variant (n = 2) and CC22-MRSA-IV + V (n = 2). The four CC97 isolates were classified as CC97-MRSA-V [fusC+] (n = 3) and CC97-MRSA-IV,WAMRSA-54/63 (n = 1). The three CC80 isolates belonged to the European CA-MRSA clone CC80-MRSA-IV [PVL+]. The three CC30 isolates belonged to three genotypes, including CC30-MRSA-IV [PVL+], Southwest Pacific clone; CC30-MRSA-IV [PVL^−/^tst1^+]^] and CC30-MRSA-IV + SCCfus [PVL+].

Two genotypes, including CC361-MRSA-IV,WAMRSA-29 (n = 2) and CC361-MRSA-V,WAMRSA-70 (n = 1), were identified among the CC361 isolates. The CC88 isolates belonged to CC88-MRSA-IV,WAMRSA-2 (n = 2) and CC88-MRSA-V [PVL+],WAMRSA-117 (n = 1). All CC121 isolates belonged to a single CC121-MRSA-V [PVL+] genotype. The two CC8 isolates were identified as ST8-MRSA-IV [PVL+/ACME+],USA300 and CC8-MRSA-V. The two CC152 isolates were identified as CC152-MRSA-V [PVL+] and CC152-MSSA [PVL+]. The latter isolate was resistant to cefoxitin, although it was identified via DNA microarray as an MSSA strain. The CC1 isolate carried the *fusC* gene and was identified as CC1-MRSA-V + SCCfus.

### 3.2. Antibiotic Resistance Phenotypes and Genotypes among MRSA Isolates

All studied MRSA isolates were susceptible to vancomycin (≤3 µg/mL), teicoplanin (≤3 µg/mL), rifampicin, linezolid (MIC ≤ 4 µg/mL), and mupirocin (≤256 µg/mL), but were resistant to penicillin (≥0.25 µg/mL) and cefoxitin (≥8 µg/mL). The MIC distributions for vancomycin were: 1.5 µg/mL (n = 9); 2 µg/mL (n = 14); and 3 µg/mL (n = 36), while the MIC distributions for teicoplanin were: 0.5 µg/mL (n = 2); 1 µg/mL (n = 2); 1.5 µg/mL (n = 14); 2 µg/mL (n = 17); and 3 µg/mL (n = 25). The isolates showed variable resistance rates to fusidic acid (n = 43, 72%), trimethoprim (n = 27, 45%), ciprofloxacin (n = 31, 51.7%), gentamicin (n = 14; 23.3%), kanamycin (n = 20; 33.3%), chloramphenicol (n = 7; 11.7%), tetracycline (n = 17; 28.3%), erythromycin (n = 19; 31.6%), inducible clindamycin (n = 13; 21.7%), and constitutive clindamycin (n = 2; 3.3%). Forty-five of the isolates were resistant to three or more classes of antibiotics and were thus classified as multi-resistant, while nine isolates were resistant to beta-lactam antibiotics only (Table 2).

All isolates were positive for *mecA*, which mediates methicillin resistance. The penicillin resistance operon *blaZ/blal/blaR* was positive in 54 isolates, while the six isolates that were phenotypically resistant to penicillin were negative. All 14 gentamicin-resistant MRSA isolates were positive for *aacA-aphD,* with four of them carrying an additional aminoglycoside resistance gene, *aadD*. All kanamycin-resistant isolates (n = 7) that were susceptible to gentamicin were positive for *aphA3* and *sat.* One phenotypically kanamycin-resistant isolate did not hybridize with the *aphA3* or *aadD*. The 19 MRSA isolates that were resistant to erythromycin carried *msrA* (n = 5), *mphC* (n = 5), and *ermC* (n = 15). All seven chloramphenicol-resistant isolates carried *fexA*. Tetracycline resistance in 17 MRSA isolates was mediated by *tet(K*) (n = 7) or *tet(M)* (n = 8). Two tetracycline-resistant isolates were negative for *tet(K)* and *tet(M)*.

Most (33/43) of the fusidic acid-resistant isolates were positive for *fusC*, while three isolates were positive for *fusB*. Seven fusidic acid-resistant isolates were negative for both *fusC* and *fusB*. Only nine of the 27 trimethoprim-resistant isolates harbored *dfrS1*, while the remaining 18 isolates did not.

### 3.3. Prevalence of Virulence Factors among MRSA Isolates

All 60 MRSA isolates carried virulence genes that code for adhesions, clumping factors A and B hemolysin, leukocidins, and biofilm formation, but varied in their carriage of genes for Panton–Valentine leukocidin (PVL), enterotoxins, accessory gene regulators (agrs), and capsular polysaccharide (cap). The distribution of the relevant virulence genes for the studied isolates are presented in Table 1. All isolates belonging to CC6, CC8, CC22, CC97, CC152, and CC361 possessed *agrI*, while *agrII* was detected in the isolates belonging to CC5 and CC15. Isolates belonging to CC1, CC30, CC80, and CC88 were positive for *agrIII*, whereas CC121 isolates were positive for *agrIV*.

All isolates were positive for either capsular polysaccharide type 5 (cap5) or type 8 (cap8). *cap5* was detected in the CC5, CC8, CC22, CC97, and CC152 isolates, while isolates of CC1, CC6, CC15, CC22, CC30, CC80, CC88, CC121, CC361 were positive for *cap8*.

Fourteen different enterotoxin-encoding genes were identified, with 50% of the isolates being positive for the enterotoxin gene cluster *egc*, comprising enterotoxins *seg, sei, selm, seln, selo, selu*. The other enterotoxin genes detected were *sea* (n = 19), *seb* (n = 15), *sed* (n = 8), *sej* (n = 6), *ser* (n = 7), *sek* (n = 4), *seq* (n = 4), and *seh* (n = 1). Panton–Valentine leucocidin (PVL) was found in 13 isolates (21.7%) belonging to CC5 (n = 1), CC8 (n = 1), CC30 (n = 2), CC80 (n = 3), CC88 (n = 1), CC121 (n = 3), and CC152 (n = 2). Toxic shock toxin-1 (*tst-1)* was detected in seven isolates belonging to CC22 (n = 4), CC30 (n = 2), and CC361 (n = 1). The gene for exfoliative toxin A (*etA*) was detected in two CC5 isolates, and the exfoliative toxin D (*etD*) was detected in three CC80 isolates.

All of the MRSA isolates carried genes for at least one of the following: staphylokinase (sak), chemotaxis inhibitory protein of *S. aureus* (*chp)*, or staphylococcal complement inhibitor (*scn*). Most of the isolates (n = 40) harbored *sak* and *scn*, while four and thirteen isolates carried *chp*/*scn* and *sak*/*chp*/*scn*, respectively. One CC6 isolate carried only *scn*. The genes for epidermal cell differentiation inhibitors type A (*edinA)* was detected in one isolate belonging to the CC5 pediatric clone, whereas the gene for epidermal cell differentiation inhibitors type B (*edinB)* was detected in isolates belonging to CC15 (n = 2) and CC80 (n = 3).

## 4. Discussion

This study has provided data on the prevalence, antibiotic resistance profile, and genotypes of MRSA isolates obtained in 2020 at the Farwaniya hospital in Kuwait. The results of this study reveal characteristic similarities and differences between the isolates obtained in this study and those obtained in the same hospital in 2016 [17]. First, there was a notable decrease in the number of isolates obtained in 2020 compared to the 209 MRSA isolates obtained in 2016 [17]. The small number of MRSA isolates obtained in 2020 could be a consequence of the COVID-19 pandemic, during which fewer patients were admitted and investigated for non-COVID-19-related cases in the hospital.

The results of the antibiotic susceptibility testing revealed that the prevalence of resistances to fusidic acid, trimethoprim, and ciprofloxacin remained high during this study, as was observed in 2016 [17]. In contrast, the proportion of gentamicin- and kanamycin-resistant isolates decreased in the present study compared to the 2016 isolates [17]. Also, the prevalence of resistances to chloramphenicol and tetracycline remained low, while resistances to erythromycin and inducible clindamycin were detected in more than 50% of the isolates, revealing similar results to those obtained in 2016 [17]. Although all of the isolates were susceptible to linezolid, rifampicin, vancomycin, and teicoplanin, the high number of isolates with vancomycin and teicoplanin MICs of 3 µg/mL is worrying, as infections caused by such isolates may not be treatable with these antibiotics. The reasons for these increases are uncertain. They probably reflect an increased usage of vancomycin and teicoplanin during the COVID-19 pandemic. These results emphasize the importance of conducting routine antibiotic susceptibility testing to monitor changes in antimicrobial resistance rates, which will help the physicians to choose the appropriate treatment and help in controlling the spread of resistant strains. Future studies should reveal whether the observed increases in the MICs of vancomycin and teicoplanin are transient or persistent.

Overall, there was high concordance between antibiotic resistance phenotypes and corresponding genotypes, except for penicillin and trimethoprim resistance. The gentamicin-resistant isolates were positive for *aacA-aphD*; the kanamycin-resistant isolates were positive for *aphA3*; the fusidic acid-resistant isolates were positive for *fusB* or *fusC*; the tetracycline-resistant isolates carried *tet(K)* or *tet(M)*; the chloramphenicol-resistant isolates carried *fexA*; and the erythromycin- and clindamycin-resistant isolates harbored *erm (C),* or *msr(A)/mph(C).* However, only 54 of the 60 penicillin-resistant isolates harbored *blaZ/blaI/blaR* and only 9 of the 27 trimethoprim-resistant isolates were positive for *dfrS1*, which mediate trimethoprim resistance in most MRSA isolates. This may be due to the presence of other determinants for resistance to penicillin G and trimethoprim [23].

Molecular typing of the MRSA isolates revealed changes in their clonal composition compared to those obtained in the same hospital in 2016. Firstly, molecular typing of the current isolates revealed three SCC*mec* types, (SCC*mec* IV, SCC*mec* V, SCC*mec* VI), thirty *spa* types, and thirteen clonal complexes (CCs). This represented a reduction in the number of SCC*mec* and *spa* types compared to the isolates investigated in the 2016 [17]. This is probably due to the smaller number of isolates investigated in the study. Secondly, the dominant *spa* types in this study were t304, t442, t688, and t1234. Whereas t304 and t688 were also common in the isolates investigated in 2016 [17], t442 is being reported here for the first time among MRSA isolates of this hospital. It is interesting to note that although t442 is emerging among MRSA isolates in Farwaniya hospital and in other hospitals in Kuwait, it has already been reported in a few human patients [24,25] and in ST5-V-MRSA isolated from chickens in India [26], suggesting that the t442 isolate belongs to a zoonotic clone. Thirdly, *spa* types t860 and t945 that were dominant in the 2016 MRSA isolates were both absent in the present study, which indicates a replacement of these *spa* types in the hospital. Both of these *spa* types are usually associated with the ST239-MRSA-III clone, which was absent in the present study. The presence of t304 and t688 both in 2016 [17] and 2020 indicates an ongoing transmission of these strains in Kuwait hospitals. Similarly, MRSA isolates belonging to t304 have also been reported to be common in Denmark, Norway, Sweden, the UK [27], Malaysia [28], and Kuwait [16,29].

The other *spa* types, t019 and t267, that occurred sporadically in this study and in the 2016 isolates [17], are common in other hospitals in Kuwait [29]. These findings illustrate the difference in the distribution of strains belonging to the same *spa* types throughout different hospitals.

The present study also revealed differences in the distribution of MRSA clonal complexes between the 2016 and 2020 isolates. Surprisingly, none of the 2020 isolates belonged to the ST239-MRSA-III clone, which was the most common MRSA clone among the 2016 Farwaniya hospital isolates [17]. Instead, CC5 and CC6 were the dominant CCs in this study. CC5, CC6, CC22, CC15, CC97, CC361, and the other sporadic isolates carried SCC*mec* types IV, V, and VI, which characterize them genotypically as community-associated MRSA (CA-MRSA) isolates. This highlights the replacement of the ST239-MRSA-III clone by CA-MRSA clones, and their expansion in the Farwaniya hospital.

The majority (n = 16) of CC5 isolates belong to the CC5-MRSA-V + SCCfus (WAMRSA-14/109) (n = 10) and CC5-MRSA-VI + SCCfus (n = 6). While CC5-MRSA-V + SCCfus (WA-MRSA-14/109) was also reported in the 2016 isolates [17] and in nearby countries [30,31], the CC5-MRSA-VI + SCCfus strain was recently identified in other hospitals in Kuwait [29] and in the United Arab Emirates [31], which illustrates the transmission of CC5-MRSA-VI + SCCfus in the Arabian Gulf region.

The CC15-MRSA reported in the present study was not detected in the Farwaniya hospital in 2016 [17]. However, CC15-MRSA isolates have been reported recently among human patients in a few hospitals in Kuwait [14]. The CC15-MRSA isolates reported in this study were resistant to gentamicin, kanamycin, fusidic acid, and tetracycline, mediated by *aacA-aphD, fusC* and *tet(K)*, respectively, and lacked genes for staphylococcal enterotoxins, similar to the CC15-MRSA isolates reported recently in other hospitals in Kuwait [14]. Therefore, the detection of CC15-MRSA in this hospital in 2020 suggests that the spread of this clone is expanding in Kuwait hospitals.

The MRSA isolates were positive for genes for different virulence factors such as Panton–Valentine leukocidin (PVL), staphylococcal enterotoxins, hemolysins, leukocidins, exfoliative toxins, epidermal cell differentiation inhibitors, biofilm production, and capsular polysaccharide. In this study, 21.7% of the isolates were positive for PVL genes. This was lower than the 30.8% prevalence of PVL-positive isolates reported in 2016 [17].

Most of the isolates in this study carried genes for multiple enterotoxins, similar to those reported previously [17]. In contrast, the proportion of isolates carrying genes for *egc* was higher in the present study than in the 2016 study.

The prevalence of genes for exfoliative toxins (*etA* and *etD*), epidermal cell differentiation inhibitors (*edinA* and *edinB*), and arginine catabolic mobile element (ACME) was low in this study, similar to the prevalence reported in the 2016 isolates [17]. The differences in the prevalence of virulence genes between this study and the previous study may be due to the differences in clonal composition encountered in both study periods.

In conclusion, this study has provided an update on the antibiotic resistance and changes in the clonal composition of MRSA in a tertiary hospital in Kuwait. It has revealed the replacement of the ST239-MRSA-III clone, which was dominant in 2016, with different CA-MRSA clones and the introduction of new strains such as CC15-MRSA and isolates with *spa* type t442 into the hospital, justifying the necessity of surveillance studies as means of contributing to the control of MRSA infections.

## Figures and Tables

**Figure 1 microorganisms-12-00017-f001:**
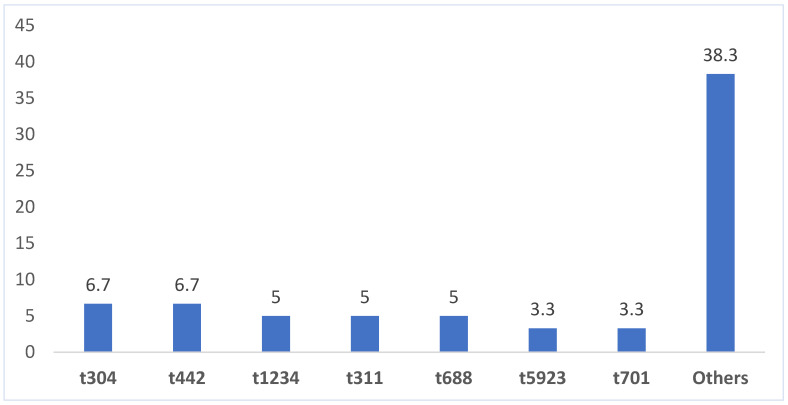
Distribution (%) of common *spa* types among MRSA isolates.

**Figure 2 microorganisms-12-00017-f002:**
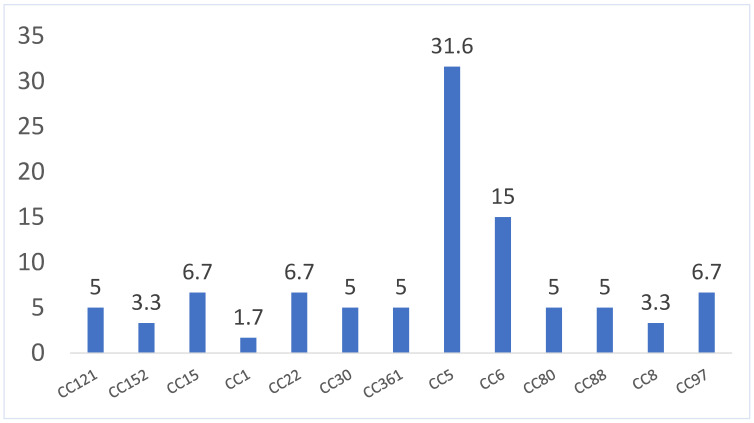
Distribution (%) of CCs among MRSA isolates.

**Table 1 microorganisms-12-00017-t001:** Molecular characteristics of MRSA isolates.

Clonal Complex (CC) (N)	Strain Type	*Spa* Type	Antibiotic Resistance Genotype	Toxins	HLB-Converting Phages	Misc. Genes
1 (1)	CC1-MRSA-V + SCCfus	ND	*blaZ,blal,blaR; aaca-aphd, aphA3; msrA,mphC; sat,fusC*	*sea,seb,seh,sek,seq*	*sak,scn*	*agrIII; cap8*
5 (19)	CC5-MRSA-IV [PVL+]	t450	*ermC*	*PVL; sea, egc*	*sak,scn*	*agrII; cap5*
	CC5-MRSA-IV, pediatric clone [edinA+],WAMRSA-65	t088	*blaZ,blal,blaR; msrA,mphC; aphA3; sat,dfrS1,tetM*	*etA; egc*	*sak,chp,scn*	*agrII; cap5; edinA*
	CC5-MRSA-V [sed/j/r+],WAMRSA-11/34/35/90/108	ND	*ermC; fusC,tetM,fexA*	*etA; sea,sed,egc,ser*	*sak,scn*	*agrII; cap5*
	CC5-MRSA-V + SCCfus,WAMRSA-14/109	t311	*blaZ,blal,blaR; ermC; fusC*	*seb,egc*	*sak,scn*	*agrII; cap5*
	CC5-MRSA-V + SCCfus,WAMRSA-14/109	ND	*blaZ,blal,blaR; fusC*	*seb,egc*	*sak,scn*	*agrII; cap5*
	CC5-MRSA-V + SCCfus,WAMRSA-14/109	t1090	*blaZ,blal,blaR; ermC; fusC*	*seb,egc*	*sak,scn*	*agrII; cap5*
	CC5-MRSA-V + SCCfus,WAMRSA-14/109	t442	*blaZ,blal,blaR; ermC; fusC*	*seb,egc*	*sak,scn*	*agrII; cap5*
	CC5-MRSA-V + SCCfus,WAMRSA-14/109	t442	*blaZ,blal,blaR; fusC*	*seb,egc*	*sak,scn*	*agrII; cap5*
	CC5-MRSA-V + SCCfus,WAMRSA-14/109	t145	*blaZ,blal,blaR; ermC; fusC*	*seb,egc*	*sak,scn*	*agrII; cap5*
	CC5-MRSA-V + SCCfus,WAMRSA-14/109	t311	*blaZ,blal; fusC*	*seb,egc*	*sak,scn*	*agrII; cap8*
	CC5-MRSA-V + SCCfus,WAMRSA-14/109	t442	*blaZ,blal,blaR; ermC; fusC*	*seb,egc*	*sak,scn*	*agrII; cap5*
	CC5-MRSA-V + SCCfus,WAMRSA-14/109	t442	*blaZ,blal,blaR; fusC*	*seb,egc*	*sak,scn*	*agrII; cap5*
	CC5-MRSA-V + SCCfus,WAMRSA-14/109	t311	*blaZ,blal,blaR; ermC; fusC*	*seb,egc*	*sak,scn*	*agrII; cap5*
PLPL	CC5-MRSA-VI + SCCfus	t232	*blaZ,blal,blaR; dfrS1,fusC,tetM,fexA*	*sea,sed,sej,egc,ser*	*sak,scn*	*agrII; cap5*
	CC5-MRSA-VI + SCCfus	t688	*blaZ,blal,blaR; dfrS1,fusC,tetM,fexA*	*sea,sed,sej,egc,ser*	*sak,scn*	*agrII; cap5*
	CC5-MRSA-VI + SCCfus	t5923	*blaZ,blal,blaR; dfrS1,fusC,tetM,fexA*	*sea,sed,sej,egc,ser*	*sak,scn*	*agrII; cap5*
	CC5-MRSA-VI + SCCfus	t688	*blaZ,blal,blaR; dfrS1,fusC,tetM,fexA*	*sed,sej,egc,ser*	*sak,scn*	*agrII; cap5*
	CC5-MRSA-VI + SCCfus	t5923	*blaZ,blal,blaR; dfrS1,fusC,tetM,fexA*	*sea,sed,sej,egc,ser*	*sak,scn*	*agrII; cap5*
	CC5-MRSA-VI + SCCfus	t688	*blaZ,blal,blaR; dfrS1,fusC,tetM,fexA,*	*sea,sed,sej,egc,ser*	*sak,scn*	*agrII; cap5*
6 (9)	CC6-MRSA-IV,WAMRSA-51	t304	*blaZ,blal,blaR; ermC*	*sea*	*scn*	*agrI; cap8*
	CC6-MRSA-IV,WAMRSA-51	t701	*ermC*	*sea*	*sak,scn*	*agrI; cap8*
	CC6-MRSA-IV,WAMRSA-51	t104	*blaZ,blal,blaR*	*sea*	*sak,scn*	*agrI; cap8*
	CC6-MRSA-IV,WAMRSA-51	t304	*blaZ,blal,blaR; ermC*		*sak,chp,scn*	*agrI; cap8*
6	CC6-MRSA-IV,WAMRSA-51	t304		*sea*	*sak,scn*	*agrI; cap8*
	CC6-MRSA-IV,WAMRSA-51	t304			*sak,scn*	*agrI; cap8*
	CC6-MRSA-IV,WAMRSA-51	t701	*blaZ,blal,blaR*	*sea*	*sak,scn*	*agrI; cap8*
	CC6-MRSA-IV,WAMRSA-51	t682	*blaZ,blal,blaR*	*sea*	*sak,scn*	*agrI; cap8*
	CC6-MRSA-IV + V,WAMRSA-66	ND	*blaZ; ermC*	*sea,sed*	*sak,scn*	*agrI; cap8*
8 (2)	CC8-MRSA-V	t650	*blaZ,blal,blaR; ermC; fusB*		*-*	*agrI; cap5*
	ST8-MRSA-IV [PVL+/ACME+],USA300	t008	*blaZ,blal,blaR; msrA,mphC; aphA3; sat*	*PVL, sek,seq*	*sak,chp,scn*	*agrI; cap5; ACME*
15 (4)	CC15-MRSA-V + SCCfus	ND	*blaZ,blal,blaR; aaca-aphd, aadD; fusC, tetK,*	-	*chp,scn*	*agrII; cap8*
	CC15-MRSA-V + SCCfus	ND	*blaZ,blal,blaR; aaca-aphd, aadD; fusC,tetK*	-	*chp,scn*	*agrII; cap8*
	CC15-MRSA-V + SCCfus	ND	*blaZ,blal,blaR; aaca-aphd, aadD; fusC,tetK,*	-	*chp,scn*	*agrII; cap8*
	CC15-MRSA-V + SCCfus	ND	*blaZ,blal,blaR; aaca-aphd, aadD; fusC,tetK*	-	*chp,scn*	*agrII; cap8*
22 (4)	CC22-MRSA-IV [tst1+], UK-EMRSA-15/Middle Eastern variant	ND	*blaZ,blal,blaR; dfrS1*	*tst1, egc*	*sak,chp,scn*	*agrI; cap5*
	CC22-MRSA-IV [tst1+], UK-EMRSA-15/Middle Eastern variant	t1120	*blaZ,blal,blaR; aacA-aphD; dfrS1*	*tst1, seb, egc*	*sak,chp,scn*	*agrI; cap5*
	CC22-MRSA-IV + V	t5168	*blaZ,blal,blaR; fusC*	*tst1, egc*	*sak,chp,scn*	*agrI; cap5;*
	CC22-MRSA-IV + V	t2336	*blaZ,blal,blaR; fucC*	*tst1, egc*	*sak,chp,scn*	*agrI; cap5*
30 (3)	CC30-MRSA-IV [PVL-/tst1+]	t391	*blaZ,blal,blaR; InuA*	*tst1, egc*	*sak,chp,scn*	*agrIII; cap8*
	CC30-MRSA-IV [PVL+], Southwest Pacific clone	t019	*blaZ,blal,blaR; msrA,mphC; aphA3; sat*	*PVL, egc*	*sak,chp,scn*	*agrIII; cap8*
	CC30-MRSA-IV + SCCfus [PVL+]	ND	*blaZ,blal,blaR; fusC*	*tst1, PVL, sea, egc*	*sak,chp,scn*	*agrIII; cap8*
80 (3)	CC80-MRSA-IV [PVL+], European caMRSA clone	t502	*blaZ,blal,blaR; aphA3; sat*	*PVL; etD*	*sak,scn*	*agrIII; cap8; edinB*
	CC80-MRSA-IV [PVL+], European caMRSA clone	t3594	*aphA3; sat*	*PVL; etD*	*sak,scn*	*agrIII; cap8; edinB*
	CC80-MRSA-IV [PVL+], European caMRSA clone	t639	*blaZ,blal,blaR; fusB*	*PVL; etD*	*sak,scn*	*agrIII; cap8; edinB*
88 (3)	CC88-MRSA-IV,WAMRSA-2	ND	*blaZ,blal,blaR; ermC*	*sea*	*sak,chp,scn*	*agrIII; cap8*
	CC88-MRSA-IV,WAMRSA-2	t3057	*blaZ,blal,blaR; ermC; tetK;*	*sea,sek,seq*	*sak, chp, scn*	*agrIII; cap8*
	CC88-MRSA-V [PVL+],WAMRSA-117	ND	*blaZ,blal,blaR; aacA-aphD*	*PVL, sea,sek,seq*	*sak,chp,scn*	*agrIII; cap8*
97 (4)	CC97-MRSA-IV,WAMRSA-54/63	t267	*blaZ,blal,blaR*		*sak,scn*	*agrI; cap5*
	CC97-MRSA-V [fusC+]	t1234	*blaZ,blal,blaR; aacA-aphD; fusC*		*-*	*agrI; cap5*
97	CC97-MRSA-V [fusC+]	t1234	*blaZ,blal,blaR; fusC*		*sak,scn*	*agrI; cap5*
	CC97-MRSA-V [fusC+]	t1234	*blaZ,blal,blaR; aacA-aphD; fusC*		*sak,scn*	*agrI; cap5*
121 (3)	CC121-MRSA-V [PVL+]	t1192	*blaZ,blal,blaR; aacA-aphD; fusC,tetK*	*PVL; seb; egc*	*sak,scn*	*agrIV; cap83*
	CC121-MRSA-V [PVL+]	t757	*blaZ,blal,blaR; aacA-aphD; fusC*	*PVL; seb; egc*	*sak,scn*	*agrIV; cap8*
	CC121-MRSA-V [PVL+]	t1991	*blaZ,blal,blaR; aacA-aphD; fusC,tetK*	*PVL; seb; egc*	*sak,scn*	*agrIV; cap8*
152 (2)	CC152-MRSA-V [PVL+]	ND	*blaZ,blal,blaR; aacA-aphD; fusC*	*PVL*	*sak,scn*	*agrI; cap5; edinB*
	CC152-MSSA [PVL+]	ND	*blaZ,blal,blaR; aacA-aphD; fucC*	*PVL*	*sak,scn*	*agrI; cap5; edinB*
361 (3)	CC361-MRSA-IV,WAMRSA-29	ND	*blaZ,blal,blaR; fusC,*	*egc*	*sak,scn*	*agrI; cap8*
	CC361-MRSA-IV,WAMRSA-29	t315	*blaZ,blal,blaR; msrA,mphC; aphA3; sat*	*egc*	*sak,scn*	*agrI; cap8*
	CC361-MRSA-V,WAMRSA-70	ND	*blaZ,blal,blaR; fusC*	*tst1, egc*	*sak,scn*	*agrI; cap5*

**Table 2 microorganisms-12-00017-t002:** Antibiotic resistance patterns of multi-resistant isolates.

No. of Antibiotic Classes	Resistance Profile	No. of Isolates
6	PG, Km, Em, Clin, Cm, Tet, Fa	1
6	PG, Km, Em, Tet, Tp, Fa	1
6	PG, Cm, Tet, Tp, Fa, Cip	3
5	PG, Gn, Km, Tet, Tp, Fa	1
5	PG, Gn, Km, Tet, Fa, Cip	1
5	PG, Gn, Km, Em, Fa, Cip	1
5	PG, Em, Clin, Tp, Fa, Cip	7
5	PG, Cm, Tet, Tp, Fa	3
5	PG, Tet, Tp, Fa, Cip	1
5	PG, Km, Tet, Fa, Cip	1
5	PG, Km, Em, Tp, Cip	2
4	PG, Em, Clin, Tet, Cip	1
4	PG, Gn, Km, Tp, Cip	1
4	PG, Gn, Km, Tet, Fa	4
4	PG, Gn, Km, Fa, Cip	2
4	PG, Tp, Fa, Cip	6
3	PG, Gn, Km, Fa	2
3	PG, Gn, Km, Cip	1
3	PG, Em, Clin, Fa	2
3	PG, Km, Fa	1
3	PG, Tp, Fa	2
3	PG, Fa, Cip	2
2	PG, Gn, Km	1
2	PG, Em, Clin	4
2	PG, Fa	2
2	PG, Cip	2
1	PG	5

**Abbreviations:** PG, penicillin G; Gn, gentamicin; Km, kanamycin; Em, erythromycin; Clin, clindamycin; Cm, chloramphenicol; Tet, tetracycline; Tp, trimethoprim; Cip, ciprofloxacin.

## Data Availability

All data are contained within the article.

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
