# Peer review of "Methicillin-Resistant Staphylococcus aureus (MRSA) in a Tertiary Care Hospital in Kuwait: A Molecular and Genetic Analysis"

_microorganisms, 2023, doi:10.3390/microorganisms12010017_

Round 1

Reviewer 1 Report

Comments and Suggestions for Authors

This manuscript describes the molecular epidemiology and antimicrobial susceptibilities of MRSA stains collected in 2020 from a tertiary care hospital in Kuwait. The study appears well performed however is entirely descriptive. The introduction states that monitoring of MRSA populations and lineages is useful to provide appropriate treatment and infection control. It would have been useful to know what actions the authors have taken or are taking on the basis of this information. In addition the antimicrobial testing of agents such as daptomycin or anti MRSA B lactams would have added to the work. However I`m not sure which of these agents are available for clinical use in Kuwait.

In more detail

lines 84-86 How were the MRSA stains collected to ensure they were representative? - sequential collection? The distribution of sites of isolation was given - is that representative of the sites of isolation at the hospital?

lines 95-97 Please give a more up to date reference to CLSI M-100 - we are now in the 33rd Edition 2023 - see reference 22 is the 2017 version. Please confirm CLSI breakpoints were used as this is not clear for vancomycin and teicoplanin (see comments below) . The BSAC clinical breakpoint for fusidic acid has been replaced by the EUCAST one S less than or equal to 1mg/L - see eucast.org clincal breakpoints and dosing v13.1 2023. Could the authors please add which agents were tested and analysed as rifampicin and linezolid appear in the Discussion but I can`t find anything about them in the Methods or Results (also see comment below)

Line 193 - not sure where the vancomycin and teicoplanin breakpoints of 3mg/L come from. As I recall both CLSI and EUCAST have clinical breakpoints of 2mg/L. Using this breakpoint then many of the MRSA isolates will be VISA which would be of great interest and of significant therapeutic and infection control importance. However testing vancomycin suscepibility of MRSA by E test can be unreliable - this needs significant clarification

Line 193-4 - one MRSA is reported to have a vancomycin MIC of 4mg/L but above it is stated all have MICs of less than or equal to 3mg/L. Please check

Lines 224-251 Inclusion of the virulence data useful - but what was the clinical, therapeutic or infection control actions as a result?

Line 268 Rifampicin and linezolid are mentioned for the first time here ? add this information to the results and methods

Author Response

The Editor -in-Chief                                                                                     7 December 2023 

Microorganisms

Re: Reviewers’ comment on manuscript: Microorganisms ID: 2742456 – Methicillin-Resistant Staphylococcus aureus (MRSA) at a tertiary care hospital in Kuwait: A molecular and genetic analysis by Wadha A. Alfouzan, Samar S.Boswihi and Edet E. Udo

Dear Sir/Madam,

Thank you for considering our paper for publication in your journal. We have read the comments by the reviewers and have responded to and incorporated suggested changes in the revised manuscript. The changes and indicated in red in the revised manuscript which we hereby submit for your necessary action.

Please find enclosed below a point-by-point response to the comments.

Response to reviewers’ comments.

Reviewer #1.

  • Comments

This manuscript describes the molecular epidemiology and antimicrobial susceptibilities of MRSA stains collected in 2020 from a tertiary care hospital in Kuwait. The study appears well performed however is entirely descriptive. The introduction states that monitoring of MRSA populations and lineages is useful to provide appropriate treatment and infection control. It would have been useful to know what actions the authors have taken or are taking on the basis of this information. In addition the antimicrobial testing of agents such as daptomycin or anti MRSA B lactams would have added to the work. However I`m not sure which of these agents are available for clinical use in Kuwait.

Response

We thank  this reviewer for his/her comments and suggestions to improve our manuscript.

  1. Regarding actions taken following our testing, we routinely transmit our typing results to Infection Control for their action.
  2. We have not yet started testing for sensitivity to daptomycin and Ceftaroline routinely in our facility.
  • Comment

lines 84-86 How were the MRSA stains collected to ensure they were representative? - sequential collection? The distribution of sites of isolation was given - is that representative of the sites of isolation at the hospital?

  • Response

The number of isolates reported was the total number obtained  in 2020. They are not representatives isolates.

  • Comment

lines 95-97 Please give a more up to date reference to CLSI M-100 - we are now in the 33rd Edition 2023 - see reference 22 is the 2017 version. Please confirm CLSI breakpoints were used as this is not clear for vancomycin and teicoplanin (see comments below) . The BSAC clinical breakpoint for fusidic acid has been replaced by the EUCAST one S less than or equal to 1mg/L - see eucast.org clincal breakpoints and dosing v13.1 2023. Could the authors please add which agents were tested and analysed as rifampicin and linezolid appear in the Discussion but I can`t find anything about them in the Methods or Results (also see comment below)

  • Response

1.We have updated the reference to  CLSI M-100 to the 30th Edition ( ref.#22)

  1. The BSAC reference has been changed to EUCAST guideline
  • Comment

Line 193 - not sure where the vancomycin and teicoplanin breakpoints of 3mg/L come from. As I recall both CLSI and EUCAST have clinical breakpoints of 2mg/L. Using this breakpoint then many of the MRSA isolates will be VISA which would be of great interest and of significant therapeutic and infection control importance. However testing vancomycin susceptibility of MRSA by E test can be unreliable - this needs significant clarification

  • Response

The vancomycin and teicoplanin MIC ≤3 µg/ml  given in the paper are  the results and not the CLSI breakpoint. The results showed that the MIC range was from 1.5 µg/ml  to 3 µg/ml  .

  • Comment

Line 193-4 - one MRSA is reported to have a vancomycin MIC of 4mg/L but above it is stated all have MICs of less than or equal to 3mg/L. Please check

  • Response

Thank you for pointing out this error. It has been corrected. The statement has been deleted.

  • Comment

Lines 224-251 Inclusion of the virulence data useful - but what was the clinical, therapeutic or infection control actions as a result?

  • Response

Virulence data maybe of interest to infection control as part of understanding the characteristic of individual clones. The relationship of carriage of virulence genes and treatment outcomes is not fully understood for now.

  • Comment

Line 268 Rifampicin and linezolid are mentioned for the first time here ? add this information to the results and methods

  • Response

The full list of antibiotics tested had been included in the methods, and results for rifampicin and linezolid is included in the results section.

Awaiting your reply

Sincerely yours,

  1. Wadha A Alfouzan, MB:BS, MSc, FRCPath

Consultant Clinical Microbiologists

Associate Professor of Clinical Microbiology

Department of Microbiology, Faculty of Medicine, Kuwait University

  1. O. Box 24923, Safat 13110

Kuwait

Tel: (+965)2498-6504

Fax: (+965)25332719

Reviewer 2 Report

Comments and Suggestions for Authors

The study addresses an important and timely issue in infectious disease control, focusing on MRSA, a major pathogen in healthcare settings.

The study offers important findings on MRSA's antibiotic resistance and clonal composition in a Kuwait hospital during 2020, using comprehensive classical and molecular methodologies. While the study adds to the understanding of MRSA in a specific setting, it has a rather geographical limited scope that somewhat constrain its impact. The authors can consider a broader comparison with similar studies across the world. The overall contribution is although significant.

Line 59: Please correct CA-MRSA

Line 89: Please delete the second word "tube"

Line 89: Please give the full name of the Vitek-2 system

Lines 146-147: Please do not capitalize biological product names

Line 331: Rephrase as "Panton-Valentine Leukocidin (PVL), other leukocidins "

Author Response

The Editor -in-Chief                                                                                     7 December 2023 

Microorganisms

Re: Reviewers’ comment on manuscript: Microorganisms ID: 2742456 – Methicillin-Resistant Staphylococcus aureus (MRSA) at a tertiary care hospital in Kuwait: A molecular and genetic analysis by Wadha A. Alfouzan, Samar S.Boswihi and Edet E. Udo

Dear Sir/Madam,

Thank you for considering our paper for publication in your journal. We have read the comments by the reviewers and have responded to and incorporated suggested changes in the revised manuscript. The changes and indicated in red in the revised manuscript which we hereby submit for your necessary action.

Please find enclosed below a point-by-point response to the comments.

Response to reviewers’ comments.

Reviewer #2

The study addresses an important and timely issue in infectious disease control, focusing on MRSA, a major pathogen in healthcare settings.

The study offers important findings on MRSA's antibiotic resistance and clonal composition in a Kuwait hospital during 2020, using comprehensive classical and molecular methodologies. While the study adds to the understanding of MRSA in a specific setting, it has a rather geographical limited scope that somewhat constrain its impact. The authors can consider a broader comparison with similar studies across the world. The overall contribution is although significant.

  • Response

We are grateful to this reviewer his/her review and for the comments to improve our paper.  On the question of a broader comparison of our findings with similar studies across the world, we have done that where appropriate. Please see lines 311-313

  • Comment

Line 59: Please correct CA-MRSA

  • Response

This has been corrected.

  • Comment

Line 89: Please delete the second word "tube"

  • Response

This has been done

  • Comment

Line 89: Please give the full name of the Vitek-2 system

  • Response

This has been given as VItek-2-Compact  (Biomerieux, USA)

  • Comment

Lines 146-147: Please do not capitalize biological product names

  • Response

This has be done

  • Comment

Line 331: Rephrase as "Panton-Valentine Leukocidin (PVL), other leukocidins "

  • Response

The statement has been rephrased. Please see line 340.

Awaiting your reply

Sincerely yours,

  1. Wadha A Alfouzan, MB:BS, MSc, FRCPath

Consultant Clinical Microbiologists

Associate Professor of Clinical Microbiology

Department of Microbiology, Faculty of Medicine, Kuwait University

  1. O. Box 24923, Safat 13110

Kuwait

Tel: (+965)2498-6504

Fax: (+965)25332719

Round 2

Reviewer 1 Report

Comments and Suggestions for Authors

None